# Evaluation of the Abbott Alinity i Thyroid-Stimulating Hormone Receptor Antibody (TRAb) Chemiluminescent Microparticle Immunoassay (CMIA)

**DOI:** 10.3390/diagnostics13162707

**Published:** 2023-08-19

**Authors:** Deborah J. W. Lee, Soon Kieng Phua, Yali Liang, Claire Chen, Tar-Choon Aw

**Affiliations:** 1Department of Laboratory Medicine, Changi General Hospital, Singapore 529889, Singapore; deborah.leejw@gmail.com (D.J.W.L.); soon_kieng_phua@cgh.com.sg (S.K.P.); yali_liang@cgh.com.sg (Y.L.); 2Abbott Laboratories, Singapore 189352, Singapore; claire.chen@abbott.com; 3Yong Loo Lin School of Medicine, National University of Singapore (NUS), Singapore 119228, Singapore; 4Duke-NUS Graduate School of Medicine, Singapore 169857, Singapore

**Keywords:** thyroid-stimulating hormone receptor antibody (TRAb), Graves’ disease, immunoassay

## Abstract

**Background:** We evaluated the performance of the Abbott thyroid-stimulating hormone receptor antibody chemiluminescent microparticle immunoassay (CMIA) on the Alinity i. **Methods:** Verification studies for precision, linearity, analytical measuring range, diagnostic cut offs for Graves’ disease were performed. We compared the Abbott CMIA to an established TRAb assay (Roche electrochemiluminescence immunoassay). Method comparison analysis was performed between serum and plasma samples on the Abbott CMIA. **Results:** Repeatability (CV%) for TRAb were 4.07, 1.56, 0.71 and within-laboratory imprecision (CV%) were 4.07, 1.90, 0.71 at 3.0, 10.0, 30.0 IU/L of TRAb, respectively. Linearity and analytical measuring range were verified from 1.07–47.9 IU/L. The limit of the blank was 0 IU/L, limit of detection was 0.15 IU/L, and limit of quantification was 0.5 IU/L. Passing-Bablok analysis showed agreement between the two assays; Y-intercept = 0.787, slope = 1.04. Passing-Bablok analysis also showed agreement between the plasma and serum samples run on the Abbott CMIA; Y-intercept −0.17, slope = 0.97. **Conclusions:** The Abbott TRAb CMIA on the Alinity i performs within the manufacturer claims for assay precision, linearity, analytical measuring range, limit of blank, limit of detection, limit of quantitation and diagnostic cut offs for Graves’ disease. Thus, the Abbott TRAb CMIA on the Alinity i is fit for clinical use.

## 1. Introduction

Thyroid receptor antibody (TRAb) measurements are useful for diagnosis as it plays a crucial role in differentiating Graves’ disease from other causes of hyperthyroidism [1,2]. Graves’ disease is the most common cause of hyperthyroidism [3]. It is an autoimmune disorder characterized by autoimmune antibodies that bind to and stimulate the thyroid-stimulating hormone receptor (TSHR) leading to an overproduction of thyroid hormones and hyperthyroid symptoms. Autoimmune antibodies that bind to the TSHR while predominantly stimulatory, may also be inhibitory, or neutral [4,5].

There are two main types of TRAb assays, bioassays and competitive immunoassays [6]. Historically, bioassays have been used to quantify stimulatory activity, where cAMP is measured consequent to TSHR stimulation. However, bioassays are labor intensive and not automated. Competitive TRAb immunoassays quantify the presence of antibodies that bind to the TSHR and do not differentiate between stimulating and blocking activity, although stimulatory antibodies predominate. There are several automated TRAb immunoassays available commercially.

This study aims to evaluate the performance of a new Abbott TRAb chemiluminescent microparticle immunoassay (CMIA) on the Alinity i platform. 

## 2. Materials and Methods

### 2.1. Materials

All serum and plasma samples used were from deidentified leftover samples stored at −70 °C, if not immediately analyzed. Frozen samples were thawed for one hour at room temperature and vortexed prior to analysis. Precision, method comparison and linearity studies were performed according to the Clinical Laboratory Standards Institute (CLSI) guidelines EP15-A3 [7], EP09c [8], EP06 [9].

For TRAb, serum is the preferred sample. Thus, 95 serum samples were analyzed on the Abbott TRAb CMIA and compared to an established TRAb assay (Roche). In addition, 88 paired serum and plasma samples were also analyzed on the Abbott TRAb CMIA to assess the effect of different matrices.

To verify the manufacturer’s diagnostic cut off for Graves’ disease (3.10 IU/L), we studied 120 healthy individuals—thyroid-stimulating hormone (TSH) 0.4–4.0 mIU/L, free thyroxine 10.0–20.0 pmol/L, and anti-thyroid peroxidase antibodies (anti-TPO) < 5.50 IU/mL (all analyzed on the Abbott Alinity i). 

### 2.2. Methods

Repeatability and within-laboratory imprecision (CV) were assessed on three levels of Abbott controls (3.0, 10.0, 30.0 IU/L). Each level of control was performed in five replicates every run, over five days. Linearity analysis was performed using samples with known high analyte concentrations which were selected to produce levels over a clinically relevant range. For the limit of the blank (LOB) and limit of detection (LOD) determination, two blank levels and two low concentration levels were run in four replicates over three days to generate 24 results per level. For the limit of quantification (LOQ) assessment, five low concentration levels were run in replicates of five over four days to generate 20 results per level. 

On the Alinity i system, the TRAb assay is an automated, delayed one-step, competitive chemiluminescent microparticle immunoassay [10]. The sample (50 μL of serum), paramagnetic microparticles coated with monoclonal mouse IgG, and assay diluent (recombinant M22-TSH receptor in HEPES buffer) are mixed and incubated in a reaction vessel on board the analyzer for approximately 20 min. Thereafter, acridinium-labelled M22-TRAb conjugate is added to the reaction vessel to complete the reaction mixture and incubated for a further 5 min. TRAb, that is present in the sample, competes with the M22-TRAb for binding to the receptor captured on the microparticles. A magnet attracts the paramagnetic particles to the wall of the reaction vessel. Following a wash cycle, unbound materials are removed. A pre-trigger solution (hydrogen peroxide) is then added to the reaction mixture to prevent any microparticle clumping and to separate the acridinium dye from conjugate bound to the microparticle complex. This is followed by addition of a trigger solution (sodium hydroxide) which causes the acridinium dye to undergo oxidation resulting in a chemiluminescent reaction. The resulting N-methylacridone that is formed releases energy (light emission) as it returns to its ground state. This chemiluminescent reaction is measured by the analyzer’s proprietary optical measurement system, where the relative light units detected have an inverse relationship to the amount of TRAb in the sample. Following a six-point calibration using the manufacturer’s materials, acceptable precision was verified with three levels of the manufacturer’s controls. From the package insert, the assay has a precision (repeatability CV%/within-laboratory CV%) of 4.8/5.2, 1.8/2.0 and 1.1/1.2 at 2.98, 9,79, 29.90 IU/L of TRAb, respectively. The limit of the blank (LOB) was 0.38 IU/L, limit of detection (LOD) was 0.62 IU/L, and limit of quantification (LOQ) was 1.06 IU/L with a linear range to 50.0 IU/L.

On the Roche Cobas e801 system, the Elecsys Anti-TSHR assay is an automated, competitive, electrochemiluminescence immunoassay (ECLIA) [11]. The sample, a pre-formed immunocomplex of solubilized TSHR and biotinylated anti-porcine TSHR mouse monoclonal antibody, and a pretreatment reagent buffer are incubated. Thereafter, buffer solution is added and further incubated. The addition of streptavidin-coated microparticles and M22 antibody labelled with a ruthenium complex compete with bound TRAb. The entire complex becomes bound to the solid phase via interaction of biotin and streptavidin. The reaction mixture is aspirated into the measuring cell where the microparticles are magnetically captured onto the surface of the electrode. Unbound substances are then removed. Application of a voltage induces chemiluminescent emission which is measured by a photomultiplier.

### 2.3. Statistical Analyses

Data were presented as means where appropriate. There were no intermediate or missing results. Passing-Bablok analysis was used to assess agreement between serum samples on the Abbott CMIA and Roche ECLIA and between serum and plasma samples on the Abbott CMIA. Bias was evaluated using the Bland–Altman method. MedCalc Statistical Software version 19.2.6 (MedCalc Software bv, Ostend, Belgium) was used for the analysis. The limit of quantification and linearity analysis were performed using Analyse-it for Microsoft Excel (version 2.30) (Analyse-it Software, Ltd., Leeds, UK). As this was part of routine clinical laboratory method evaluation using deidentified leftover sera, national regulations exempt such investigations from Institutional Review Board (IRB) review.

## 3. Results

### 3.1. Performance

Repeatability and within-laboratory precision, calculated by five-day analysis of three levels of control materials run in replicates of five are reported in Table 1. The repeatability CV for the three levels of control were 4.07, 1.56 and 0.71%, respectively. The within-laboratory CVs were 4.07, 1.90 and 0.71%, respectively. These were lower than the manufacturer claimed CVs at each level. 

Linearity for the TRAb CMIA is shown in Figure 1. Although a polynomial fit is better than a linear fit, the deviation from a linear fit is not significant (*p* < 0.0001). The analytical measuring interval was determined to be 1.07–47.9 IU/L. All blank samples returned a value of 0 IU/L and all low-concentration samples (0.59–0.81 IU/L) returned a non-zero value; LOD = 0.15 IU/L. The LOQ was verified with samples ranging from 0–2.7 IU/L, where the CV at 1.06 IU/L was 9.9%; LOQ = 0.5 IU/L. Of the 120 healthy patient samples assayed, TRAb results ranged from 0.36–1.94 IU/L (100% of results below the given manufacturer diagnostic cut off 3.10 IU/L for Graves’ disease).

### 3.2. Method Comparison

#### 3.2.1. Abbott CMIA and Roche CLIA Method Comparison

Method comparison was performed on 95 serum samples, covering a wide range from <1.1 to >40.0 IU/L (Roche ECLIA) and 0.71 to >50.0 IU/L (Abbott CMIA). Only results within the measuring range of both assays (*n* = 69) were assessed with Passing-Bablok and Bland–Altman analyses. The y-intercept was 0.787, slope was 1.04 and the 95% confidence intervals (CIs) were 0.25 to 1.33 and 0.93 to 1.13, respectively. A Cusum test was not significant for deviation from linearity (*p* = 0.29), and the Spearman rank correlation coefficient was 0.95 (95% CI 0.93 to 0.97) with *p* < 0.0001. The Abbott CMIA had a persistent positive bias of 0.79 IU/L (relative bias: 16.5%) compared to the Roche ECLIA. Results are shown in Figure 2.

Concordance analysis was performed according to the manufacturer-determined diagnostic cut offs for Graves’ disease on the 95 serum samples. The cut offs were 3.10 IU/L for the Abbott CMIA and 1.75 IU/L for the Roche ECLIA. The results were classified as reactive or non-reactive. There was agreement (94.7%) between the results of the two assays. Discordant results (*n* = 5) bordered the decision limits (Roche ECLIA: 2.0–3.3, Abbott CMIA: 0.86–2.54 IU/L). All discordant results were reactive on the Roche ECLIA and non-reactive on the Abbott CMIA.

#### 3.2.2. Abbott Serum and Plasma Method Comparison

Method comparison was performed with 88 paired serum and plasma samples on the Abbott CMIA, covering a wide range from 0.63 to >50.0 IU/L (serum) and 0.53 to >50.0 IU/L (plasma). Only results within the measuring range (*n* = 86) were assessed with Passing-Bablok and Bland–Altman analyses. The y-intercept was −0.17, the slope was 0.97 and the 95% CIs were −0.27 to −0.05 and 0.95 to 1.00, respectively. Cusum test was not significant for deviation from linearity (*p* = 0.78) and spearman rank correlation coefficient was 0.97 (95% CI 0.95 to 0.98) with *p* < 0.0001. The plasma samples had a persistent negative bias of 0.32 IU/L (relative bias: 11.6%) compared to serum samples. Results are shown in Figure 3.

Concordance analysis was performed using the manufacturer diagnostic cut off for Graves’ disease (3.10 IU/L) on the 88 paired serum and plasma samples and classified as reactive or non-reactive. There was agreement (97.7%) between the results from serum and plasma specimens. Discordant results (*n* = 2) bordered the decision limit (serum: 3.21–3.53, plasma: 2.73–2.83 IU/L). All discordant results were reactive on serum and non-reactive on plasma.

## 4. Discussion

TRAbs are the diagnostic marker for Graves’ disease, and monitoring pretreatment levels and levels before ceasing therapy provides valuable prognostic information [12]. High pretreatment TRAb levels are associated with less response to anti-thyroid drugs and higher rates of disease recurrence [13] as well as a risk of developing Graves’ ophthalmopathy [14,15,16]. TRAb levels are measured before cessation of treatment because patterns in TRAb changes can predict the risk of recurrence and guide further management [4,17,18]. The presence of TRAbs can also indicate a risk of Graves’ disease in patients with subclinical hypothyroidism, and predict fetal and neonatal thyrotoxicosis [6]. The current TRAb assays are 3rd generation. In our laboratory, we had previously compared the 3rd generation Roche ECLIA with the 2nd generation Brahms TSHR antibody concentration (TRAK) radio-receptor assay. This study aims to compare our current 3rd generation Roche ECLIA with the new 3rd generation Abbott Alinity i CMIA. 

Our study verified that the Abbott TRAb CMIA shows good performance and is in agreement with the manufacturer’s claims. Our evaluated precision was similar to a previously reported study comparing the Abbott TRAb and Roche Cobas e411 TRAb assays [19]. In that study, LOQ was verified with CV 8.4% at 1.22 IU/L. Our CV at the manufacturer’s claimed LOQ (1.06 IU/L) was 9.9%, and the measured functional sensitivity was 0.50 IU/L. Their study similarly found that all serum samples from 187 apparently healthy patients (no prescribed medications, normal TSH and free T4) had TRAb measurements less than the claimed 3.10 IU/L cut off for differentiating Graves’ disease from other types of hyperthyroidism. Our study is the first to compare the Abbott Alinity i TRAb to the Roche Cobas e801 TRAb assay, and both instruments show close agreement. 

Serum is the only recommended specimen type by both Roche and Abbott for their TRAb assays [10,11]. As our laboratory accepts plasma specimens for other commonly ordered thyroid tests (e.g., TSH, free thyroxine, anti-TPO antibodies, anti-thyroglobulin antibodies), we decided to assess the comparability of plasma and serum specimens. In limited studies we previously performed tests with the Roche ECLIA, and some normal serum samples became reactive on paired plasma samples. However, this is not the case with the Abbott CMIA, as shown. In the Abbott CMIA, there was close agreement between plasma and serum samples using Passing-Bablok analysis (97.7% concordance). Plasma showed a minimal persistent negative bias of 0.32 IU/L (relative bias: 11.6%) compared to serum over a serum concentration from 0.63–35.4 IU/L. Our study suggests that plasma specimens are an acceptable sample type for the Abbott CMIA. Further studies are needed to confirm the use of plasma specimens for the Abbott TRAb assay.

The Abbott Alinity i TRAb CMIA is a newly available commercial TRAb assay which has some differences from the Roche Cobas Elecsys Anti-TSHR assay. These are summarized in Table 2.

In addition to these differences, the Roche ECLIA is a biotinylated assay which may be susceptible to biotin interference at extremely high serum levels (>600 ng/L) [11] and in those with renal failure [20]. The advantage of the Abbott CMIA is evident from Table 2. All reagents, calibrators and controls are ready to use without the need for pretreatment as in the Roche method. The Abbott assay calibration curve is stable for 7 days unlike the Roche procedure which requires daily calibration. Abbott employs a six-point calibration instead of two (on the Roche), thus giving greater confidence in the signal-to-concentration relationship. The Abbott protocol provides three controls which span a wider concentration range (30 IU/L) rather than only two for the Roche. The Abbott assay provides onboard auto-dilution for samples with high concentration (>50 IU/L) rather than manual dilution for Roche on samples >40 IU/L. 

A limitation of our study is that we were unable to verify the interferences by anti-thyroglobulin antibodies, anti-TPO, follicle stimulating hormone, human chorionic gonadotropin, IgG, IgM, luteinizing hormone, and TSH declared by the manufacturer. As the stated thresholds for cross-reaction are quite high, samples with those concentrations are expected to be rare. 

## 5. Conclusions

This study verified that the Abbott TRAb CMIA on the Alinity i performs within the manufacturer’s claims for assay precision, linearity, analytical measuring range, limit of the blank, limit of detection, limit of quantitation and diagnostic cut offs for Graves’ disease. Our results provide independent verification that the Abbott assay compares very favorably to an established 15-year-old Roche assay on their main immunoassay platform the Cobas e801. Thus, the Abbott TRAb CMIA on the Alinity i is fit for clinical use. 

## Figures and Tables

**Figure 1 diagnostics-13-02707-f001:**
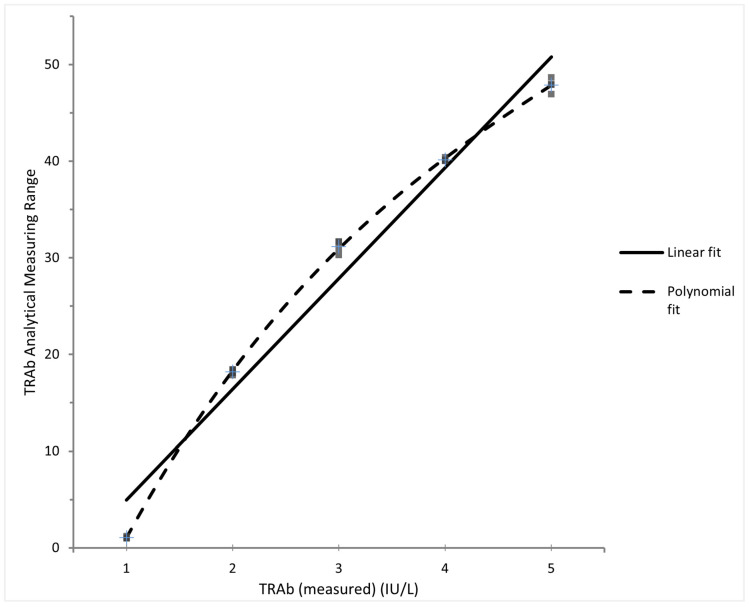
TRAb linearity results plotted over the manufacturer-declared analytical measuring range.

**Figure 2 diagnostics-13-02707-f002:**
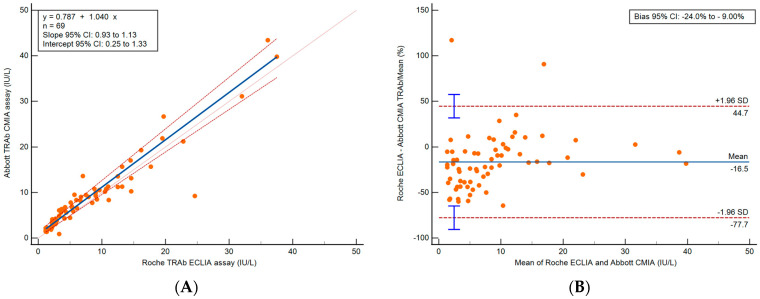
Method comparison results for (*n* = 69) TRAb (IU/L) for samples within the measuring range only. (**A**) Passing-Bablok regression and (**B**) Bland–Altman analysis of Roche ECLIA vs. Abbott CMIA.

**Figure 3 diagnostics-13-02707-f003:**
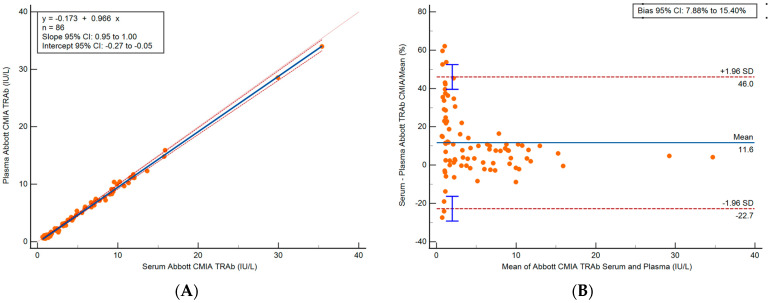
Method comparison results for (*n* = 86) TRAb (IU/L) for samples within the measuring range only. (**A**) Passing-Bablok regression and (**B**) Bland–Altman analysis of serum vs. plasma samples on the Abbott CMIA.

**Table 1 diagnostics-13-02707-t001:** Precision results for TRAb assay expressed as imprecision (CV) in percentage (%), obtained using three levels of Abbott controls.

Measurand	Level (IU/L)	Design	Measured Repeatability, CV%	Manufacturer Claimed Repeatability, CV%	Measured Within-Laboratory Imprecision, CV%	Manufacturer Claimed Within-Laboratory Imprecision, CV%
TRAb	3.0	5 × 5 CLSI EP15-A3	4.07	4.8	4.07	5.2
	10.0		1.56	1.8	1.90	2.0
	30.0		0.71	1.1	0.71	1.2

Abbreviations CV: imprecision; TRAb: thyroid receptor antibody; CLSI: Clinical and Laboratory Standards Institute.

**Table 2 diagnostics-13-02707-t002:** Differences between Abbott Alinity i TRAb CMIA and Roche Cobas Elecsys Anti-TSHR ECLIA.

Characteristics	Abbott Alinity i TRAb CMIA	Roche Cobas Elecsys Anti-TSHR ECLIA
**Sample volume**	100 µL	30 µL
**Assay time**	29 min	27 min
**Reference standard**	NIBSC 2nd IS 08/204	NIBSC 1st IS 90/672
**Reagent, calibrator, and** **control preparation**	Nil (ready-to-use)	Pretreatment required
**Reagent onboard stability**	7 days	16 weeks (with daily calibration)
**Calibration**	Six-point calibration	Two-point calibration
**Controls**	3.0, 10.0, 30.0 IU/L	4.0, 10.0 IU/L
**Dilution of high samples**	Auto-dilution (1:10 for samples > 50 IU/L)	Manual dilution (1:10 for samples > 40 IU/L)

Abbreviations: TRAb: thyroid receptor antibody; anti-TSHR: anti-thyroid-stimulating hormone receptor; NIBSC: National Institute for Biological Standards and Control; IS: international standard.

## Data Availability

The datasets generated during and/or analyzed during the current study are not publicly available due to privacy issues and national laws but are available from the corresponding author on reasonable request under the provision that data may not leave the hospital/center premises.

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
