# Peer review of "Evaluation of the Abbott Alinity i Thyroid-Stimulating Hormone Receptor Antibody (TRAb) Chemiluminescent Microparticle Immunoassay (CMIA)"

_diagnostics, 2023, doi:10.3390/diagnostics13162707_

Round 1
Reviewer 1 Report
"Evaluation of the Abbott Alinity i Thyroid Stimulating Hormone Receptor Antibody (TRAb) Chemiluminescent microparticle immunoassay (CMIA)" by Deborah J. W. Lee , Soon Kieng Phua , Yali Liang , Claire Chen , Tar-Choon Aw compares the two techniques well. The number of repetitions and different time intervals allow you to compare techniques. It is important that for the development of the method on real samples, parameters such as the limit value of the baseline were taken into account. Even though sometimes the errors are quite high, this has very little effect on the result. The manuscript is clearly written.
Author Response
"Evaluation of the Abbott Alinity i Thyroid Stimulating Hormone Receptor Antibody (TRAb) Chemiluminescent microparticle immunoassay (CMIA)" by Deborah J. W. Lee , Soon Kieng Phua , Yali Liang , Claire Chen , Tar-Choon Aw compares the two techniques well. The number of repetitions and different time intervals allow you to compare techniques. It is important that for the development of the method on real samples, parameters such as the limit value of the baseline were taken into account. Even though sometimes the errors are quite high, this has very little effect on the result. The manuscript is clearly written.
Response: We thank reviewer 1 for the comments and we agree. We have independently verified the LoB, LoD and LoQ (line 122-125) as the stated values are those provided by the manufacturer (line 83-84).
==================================
Reviewer 2 Report
The CLSI guidelines referenced by the authors for linearity verification evaluation were previous versions and the calculations were not accurately presented. I recommend entering it in the new version, EP06-Ed2 workbook Excel file. It is easier to verify with only 5 to 6 repeat measurement data.
line 77 acceptable performance -> acceptable precision or imprecision
Author Response
The CLSI guidelines referenced by the authors for linearity verification evaluation were previous versions and the calculations were not accurately presented. I recommend entering it in the new version, EP06-Ed2 workbook Excel file. It is easier to verify with only 5 to 6 repeat measurement data.
line 77 acceptable performance -> acceptable precision or imprecision
Response: We thank reviewer 2 for the comments and suggestions. We included the polynomial and linear fit for the benefit of readers who might still be used to the previous EP06. We calculated the linear fit displayed as a solid line intentionally in Figure 1; we demonstrated that there was no significant difference between the two as we stated in line 120-121. However we have corrected reference 9 to reflect the current edition of EP06. (line 271-272).
==================================
Reviewer 3 Report
The article is about a newly available commercial TRAb assay. In this work, blood plasma was analyzed using Abbott CMIA (according to the manufacturer's instructions) and compared with a similar analysis. It is not clear what is the novelty of the study? What is the advantage of Abbott CMIA analysis?
The methods are not described clearly and in sufficient detail. All stages of the experiment should be described in detail: how much and at what concentration the sample, microparticles, diluent, etc. were added. Lines 83-92 are not descriptions of the method, this should be in the results and discussion section. What is the meaning of ruthenium complex? Which photomultiplier was used?
The obtained results do not bear any scientific significance. The conclusions duplicate the conclusion of the manufacturer.
Author Response
The article is about a newly available commercial TRAb assay. In this work, blood plasma was analyzed using Abbott CMIA (according to the manufacturer's instructions) and compared with a similar analysis. It is not clear what is the novelty of the study? What is the advantage of Abbott CMIA analysis?
Response: It is the responsibility of all Clinical Laboratories to evaluate all newly introduced assays for comparability and to see their fitness for purpose. That is why we undertook this investigation. There is no previous study that we are aware that has compared the Abbott TRAb to the established Roche TRAb save Choksi et al (ref 19) but it was on the small Roche e411.
The advantage of the Abbott CMIA is evident from Table 2. All reagents, calibrators and controls are ready to use without the need for pre-treatment as in the Roche. The Abbott assay calibration curve is stable for 7 days unlike the Roche method which requires daily calibration. The Abbott uses a 6-point calibration instead of two (on the Roche) giving greater confidence in the signal to concentration relationship. The Abbott method provides 3 controls which span a wider concentration range (30IU/L) rather than only 2 for the Roche. The Abbott assay provides on-board autodilution for samples with high concentration (>50IU/L) rather than manual dilution for Roche on samples >40IU/L. Besides the Abbott assay is not subject to biotin interference as it does not employ streptavidin-biotin in the assay unlike Roche.
The methods are not described clearly and in sufficient detail. All stages of the experiment should be described in detail: how much and at what concentration the sample, microparticles, diluent, etc. were added. Lines 83-92 are not descriptions of the method, this should be in the results and discussion section. What is the meaning of ruthenium complex? Which photomultiplier was used?
Response: We have provided sufficient detail in the Methods section (lines 62-95) including abridged descriptions of both the Abbott & Roche chemiluminescent assays as a convenient reminder for readers new to laboratory immunoassays. Details are also available in the manufacturers’ instructions for use (reference 10 & 11).
We beg to differ from reviewer 3 with respect to “lines 83-92”. It will be evident from a careful reading of the new lines 85-95 that the section is indeed related to methods. Thus we have left it iuncorrected.
Chemiluminescence immunoassays have now been available for over 25 years [Kricka Luminescence 1999;14(2)113-118] and Clinical laboratories are very familiar with their use and need no elaboration of ruthenium and photomultiplier.
The obtained results do not bear any scientific significance. The conclusions duplicate the conclusion of the manufacturer.
Response: These results are very significant. It provides independent verification that the Abbott assay compares very favorably to an established 15-year old Roche assay on their main immunoassay platform the e801. Besides it has also many advantages as shown in Table 2. Our conclusions are what will help other laboratorians – that the Abbott assay is fit for purpose when they consider introducing a TRAB test or are considering replacing their existing one.
==================================
Reviewer 4 Report
Lee et al evaluated the performance of the Abbott thyroid stimulating hormone 13 receptor antibody chemiluminescent microparticle immunoassay (CMIA) on the Alinity i. moreover, they compared the Abbott CMIA to an established TRAb assay 16 (Roche electrochemiluminescence immunoassay), in serum and plasma. The study is well structured and the results are clear. I have only few observation:
Introduction:
Line 43-33: There are several automated competitive 43 immunoassays available commercially. Please explain.
Materials and methods:
It would be clearer if divided into sub-sections, for example: study population, methods, statistical analysis...
Discussion
Line 172-174
In our laboratory, we previously compared the 3rd 172 generation Roche ECLIA with the 2nd generation Brahms TSHR antibody concentration 173 (TRAK)radio-receptor assay. Please add the reference or better explain.
Author Response
Lee et al evaluated the performance of the Abbott thyroid stimulating hormone 13 receptor antibody chemiluminescent microparticle immunoassay (CMIA) on the Alinity i. moreover, they compared the Abbott CMIA to an established TRAb assay 16 (Roche electrochemiluminescence immunoassay), in serum and plasma. The study is well structured and the results are clear. I have only few observation:
Introduction:
Line 43-33: There are several automated competitive 43 immunoassays available commercially. Please explain.
Response: What we wanted to say is that there are several automated TRAb immunoassays available commercially e.g. RSR, Roche, Siemens, SNIBE besides Abbott to name a few. We have thus corrected the sentence accordingly
Materials and methods:
It would be clearer if divided into sub-sections, for example: study population, methods, statistical analysis...
Response: We thank reviewer 4 for the suggestion. Corrected as suggested.
Discussion
Line 172-174
In our laboratory, we previously compared the 3rd 172 generation Roche ECLIA with the 2nd generation Brahms TSHR antibody concentration 173 (TRAK)radio-receptor assay. Please add the reference or better explain.
Response: We evaluated the Roche TRAb ECLIA on the Cobas e601 in 2007/2008 and presented it at the 2008 AACC meeting: Aw TC, Wong PW, Phua SK, Tan SP. Performance of a new chemiluminescent TSH receptor antibody assay. Clinical Chemistry 2008 Supplement Vol 55 (S6): abstract C54. However the online abstracts of AACC meetings only date back to 2009. Thus we have no verifiable online media other than hardcopy in the library to cite this.
==================================
Round 2
Reviewer 3 Report
The new version of the article slightly differs from the previous one. Most of the comments have not been corrected.
The methods of analysis are not described in sufficient detail. The volume of plasma samples introduced, the amount of reagents introduced and the incubation time are missing. There is no detailed description of the washing steps. The incubation time is not indicated for each stage. There is also no method for measuring the chemiluminescent reaction.
The novelty of the research should be reflected in the article, not only in the coverletter for the reviewer. The authors indicate that the benefits are listed in Table 2, but these benefits should be described in the text.
Author Response
The manuscript has been revised as you have suggested and is attached.
The methods section of the previous manuscript line 72-79 has been expanded (line 72-89) to include as much detail as we can glean from the package insert and operator instrument manual. Unlike lab developed tests in the realm of clinical laboratory analyses what you see is what you get.
As requested we have described the advantages and benefits of the new Abbott assay (line 220-228). In addition we have also highlighted it in the Conclusion section (line 238-240).
==================================
Round 3
Reviewer 3 Report
The quality of the paper had improved, and all my questions were addressed. No more comments.